# Evaluating the effect of metabolic traits on oral and oropharyngeal cancer risk using Mendelian randomization

Mark Gormley[1,2]*, Tom Dudding[2], Steven J Thomas[2], Jessica Tyrrell[3], Andrew R Ness[4], Miranda Pring[2], Danny Legge[5], George Davey Smith[1], Rebecca C Richmond[1], Emma E Vincent[1,5†], Caroline Bull[1,5†]

[1]MRC Integrative Epidemiology Unit, Population Health Sciences, Bristol Medical School, University of Bristol, Bristol, United Kingdom; [2]Bristol Dental Hospital and School, University of Bristol, Bristol, United Kingdom; [3]University of Exeter Medical School, RILD Building, RD&E Hospital, Exeter, United Kingdom; [4]University Hospitals Bristol and Weston NHS Foundation Trust National Institute for Health Research Bristol Biomedical Research Centre, University of Bristol, Bristol, United Kingdom; [5]Translational Health Sciences, Bristol Medical School, University of Bristol, Bristol, United Kingdom

*For correspondence: mark.gormley@bristol.ac.uk

†These authors contributed equally to this work

Competing interest: The authors declare that no competing interests exist.

**Abstract** A recent World Health Organization report states that at least 40% of all cancer cases may be preventable, with smoking, alcohol consumption, and obesity identified as three of the most important modifiable lifestyle factors. Given the significant decline in smoking rates, particularly within developed countries, other potentially modifiable risk factors for head and neck cancer warrant investigation. Obesity and related metabolic disorders such as type 2 diabetes (T2D) and hypertension have been associated with head and neck cancer risk in multiple observational studies. However, adiposity has also been correlated with smoking, with bias, confounding or reverse causality possibly explaining these findings. To overcome the challenges of observational studies, we conducted two-sample Mendelian randomization (inverse variance weighted [IVW] method) using genetic variants which were robustly associated with adiposity, glycaemic and blood pressure traits in genome-wide association studies (GWAS). Outcome data were taken from the largest available GWAS of 6034 oral and oropharyngeal cases, with 6585 controls. We found limited evidence of a causal effect of genetically proxied body mass index (BMI; OR IVW = 0.89, 95% CI 0.72–1.09, p = 0.26 per 1 standard deviation in BMI [4.81kg/m$^2$]) on oral and oropharyngeal cancer risk. Similarly, there was limited evidence for related traits including T2D and hypertension. Small effects cannot be excluded given the lack of power to detect them in currently available GWAS.

## Editor's evaluation

This work presents valuable findings on the causal association of metabolic traits and head and neck cancers. The evidence supporting the conclusion is convincing, with rigorous and comprehensive data analysis. The work will be of interest to cancer epidemiologists, especially those working on head and neck cancer.

## Introduction

Head and neck squamous cell carcinoma (HNC), which includes oral and oropharyngeal cancer is the seventh most common cancer, accounting for more than 660,000 new cases and 325,000 deaths

annually worldwide (*Johnson et al., 2020*; *Sung et al., 2021*). Established risks include tobacco use, alcohol consumption (*Hashibe et al., 2009*), and human papillomavirus (HPV) infection, mainly associated with oropharyngeal cancer and thought to be sexually transmitted (*Gillison et al., 2015*). A recent World Health Organization (WHO) report states that at least 40% of all cancer cases may be preventable, with smoking, alcohol consumption, and obesity identified as three of the most important modifiable lifestyle factors (*World Health Organization, 2022*). Smoking behaviour is declining, particularly in developed countries (*Dai et al., 2022*) and it has been projected that obesity could even supersede smoking as the primary driver of cancer in the coming decades (*World Health Organization, 2022*). Despite changes in smoking rates, the incidence of HNC continues to rise and a changing aetiology has been proposed (*Conway et al., 2018*; *Thomas et al., 2018*). Therefore, less established risks such as obesity and its related metabolic traits warrant investigation in HNC. However, obesity has been correlated with other HNC risk factors such as smoking (*Carreras-Torres et al., 2018*), alcohol (*Carter et al., 2019a*) and educational attainment (*Carter et al., 2019b*), meaning independent effects are difficult to establish.

Obesity is now considered to increase the risk of at least 13 different types of cancer including breast, colorectal, gastric, and oesophageal (*Centers for Disease Control and Prevention, 2021*), but the effect on HNC risk remains unclear [*World Health Organization, 2022*]. Public health strategies have been unsuccessful in addressing the current obesity epidemic at the population level, which could result in more cancer cases in the years to come (*Davey, 2004*). Obesity and related metabolic traits such as type 2 diabetes (T2D), hypertension, and dyslipidaemia have all been associated with HNC in multiple observational studies. In the largest pooled analysis, obesity defined by higher body mass index (BMI) was associated with a protective effect for HNC in current smokers (hazard ratio [HR]0.76, 95% confidence intervals [95% CI] 0.71–0.82, p<0.0001, per 5 kg/m$^2$) and conversely, a higher risk in never smokers (HR 1.15, 95% CI 1.06–1.24 per 5 kg/m$^2$, p < 0.001) (*Gaudet et al., 2015*). In the same study, a greater waist circumference (WC) (HR 1.04, 95% CI 1.03–1.05 per 5 cm, p < 0.001) and waist-to-hip ratio (WHR) (HR 1.07, 95% CI 1.05–1.09 per 0.1 unit, p < 0.001) were associated with increased HNC risk, which did not vary by smoking status (*Gaudet et al., 2015*). However, more recent cohort studies have failed to show a clear association between BMI and HNC (*Cao et al., 2020*; *Recalde et al., 2021*; *Gribsholt et al., 2020*; *Jiang et al., 2021*; *Ward et al., 2017*). A random-effects meta-analysis of observational studies showed an increased association between T2D and oral

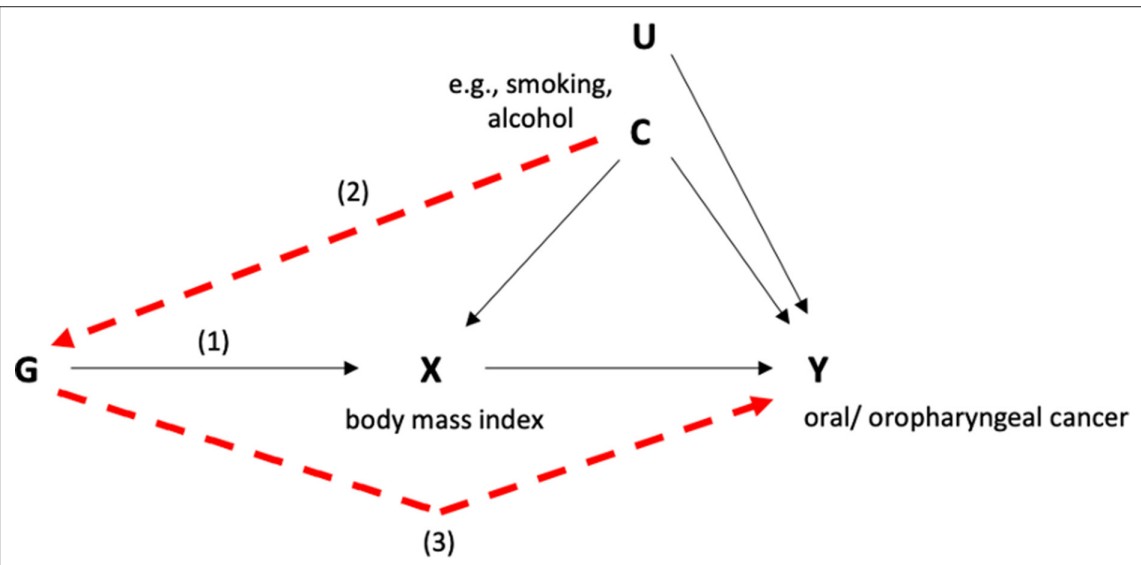

**Figure 1.** Directed acyclic graph (DAG) depicting Mendelian randomization applied to this study. Genetic variants (G) can act as proxies or instruments to investigate if an exposure (X) is associated with a disease outcome (Y). Causal inference can be made between X and Y if the following conditions are upheld. (1) The genetic variants which make up the instrument are valid and reliably associated with the exposure (i.e., the 'relevance assumption'); (2) There is no measured or unmeasured confounding of the association between the genetic instrument and the outcome (i.e., the 'exchangeability' assumption); (3) There is no independent pathway between the genetic instrument and the outcome, except through the exposure (i.e., the 'exclusion restriction principle').

and oropharyngeal cancer (risk ratio [RR] of 1.15, 95% CI 1.02–1.29, p < 0.001 [*Gong et al., 2015*]). This result is consistent with more recent independent cohorts (*Jiang et al., 2021*; *Kim et al., 2021*; *Kim et al., 2019*; *Saarela et al., 2019*). Hypertension (defined as a systolic blood pressure [SBP] >130 mmHg or diastolic blood pressure [DBP] >85 mmHg) has been correlated with HNC risk across multiple studies (*Christakoudi et al., 2020*; *Kim et al., 2021*; *Kim et al., 2019*; *Seo et al., 2020*; *Stocks et al., 2012*). Nonetheless, selection bias, confounding, or reverse causation may explain the findings from these studies.

Mendelian randomization (MR) is an analytical approach which attempts to overcome the challenges of conventional epidemiological studies. The method uses germline genetic single nucleotide polymorphisms (SNPs), which are randomly assorted during meiosis (and fixed at conception), to estimate the causal effects of exposures on disease outcomes (*Smith and Ebrahim, 2003*; *Davey Smith and Hemani, 2014*; *Sanderson et al., 2022*). MR makes three key assumptions, as described in *Figure 1* (*Smith and Ebrahim, 2003*; *Davey Smith and Hemani, 2014*). To instrument metabolic traits, we selected genetic variants associated ($p < 5 \times 10^{-8}$) with traits of interest identified by previously conducted genome-wide association studies (GWAS; *Supplementary file 1*). Further detail on MR methods and sensitivity analyses is given in the Materials and methods.

Using MR, we recently found limited evidence for a role of circulating lipid traits in oral and oropharyngeal cancer risk (*Gormley et al., 2021*), however other metabolic traits remain untested in an MR framework. This study aims to examine the causal effect of metabolic traits on the risk of oral and oropharyngeal cancer using two-sample MR. Specifically, we will examine adiposity measures (BMI, WC, WHR), glycaemic traits (T2D, glycated haemoglobin [HbA$_{1c}$], fasting glucose [FG], fasting insulin [FI]), and blood pressure (SBP, DBP). Given the potential correlation of metabolic traits and established HNC risk factors, further evaluation of instrument-risk factor effects including smoking, alcohol, risk tolerance (as a proxy for sexual behaviour), and educational attainment was carried out using MR.

## Results

*F*-statistics of genetic instruments for metabolic traits ranged from 33.3 to 133.6, indicating sufficient instrument strength for MR analyses (*Supplementary file 2*, Table 2A). Genetic instruments were estimated to explain between 0.5% (FI) and 4% (BMI) of their respective metabolic trait (*Supplementary file 2*, Table 2A). Based on the results of prior observational studies we would expect to detect OR of >1.2 for a clinically meaningful effect of metabolic traits on oral and oropharyngeal cancer. *Figure 2—figure supplement 1* displays power estimates for MR analyses. In analyses where BMI was the exposure, we had 80% power to detect an association with an OR of 1.2 or more at an $\alpha$ of 0.05 for combined oral and oropharyngeal cancer. Power was lower for other metabolic traits and reduced when stratifying analyses by subsite.

### Estimated effect of adiposity on oral and oropharyngeal cancer risk

There was limited evidence of an effect of higher BMI or WHR on combined oral and oropharyngeal cancer (OR IVW = 0.89, 95% CI 0.72–1.09, p = 0.26, per 1 standard deviation [SD] in BMI [4.81 kg/m$^2$] and OR IVW = 0.98, 95% CI 0.74–1.29, p = 0.88, per 1 SD in WHR [0.10 unit]) (*Table 1*, *Figure 2*, *Figure 2—figure supplements 2 and 3*). Results were consistent when analyses were stratified by subsite (*Table 1*). WC, another measure of adiposity did show a protective direction of effect (OR IVW = 0.73, 95% CI 0.52–1.02, p = 0.07, per 1 SD increase in WC [0.09 unit]), particularly in the oropharyngeal subsite (OR IVW = 0.66, 95% CI 0.43–1.01, p = 0.06, per 1 SD increase in WC [0.09 unit]) (*Table 1*, *Figure 2*, *Figure 2—figure supplement 4*).

### Estimated effect of glycaemic traits on oral and oropharyngeal cancer risk

There was limited evidence for an effect of genetically proxied T2D on combined oral and oropharyngeal cancer [OR IVW = 0.92, 95% CI 0.84–1.01, p = 0.09, per 1-log unit higher odds of T2D (*Table 1*, *Figure 2*, *Figure 2—figure supplement 5*)]. Traits related to diabetes, including HbA$_{1c}$ resulted in a weak protective effect on combined oral and oropharyngeal cancer risk (OR IVW = 0.56, 95% CI 0.32–1.00, p = 0.05, per 1-log unit % higher HbA$_{1c}$), which remained only in the oral subsite (OR IVW = 0.48, 95% CI 0.24–0.93, p = 0.03, per 1-log unit % higher HbA$_{1c}$) following stratification (*Table 1*,

**Table 1.** Mendelian randomization results of genetically proxied metabolic traits with risk of oral and oropharyngeal cancer in GAME-ON.

| Exposure | Outcome | Exposure/outcome source | Outcome N | Number of SNPs | IVW OR (95% CI) | p | Weighted median OR (95% CI) | p | Weighted mode OR (95% CI) | p | MR-Egger OR (95% CI) | p |
|---|---|---|---|---|---|---|---|---|---|---|---|---|
| BMI | Oral and oropharyngeal cancer combined | | 6034 | 272 | 0.89 (0.72, 1.09) | 0.26 | 0.71 (0.50, 1.00) | 0.05 | 0.63 (0.37, 1.04) | 0.07 | 0.66 (0.40, 1.10) | 0.11 |
| | Oral cancer | | 2990 | 272 | 0.92 (0.71, 1.19) | 0.53 | 0.83 (0.55, 1.28) | 0.40 | 0.79 (0.38, 1.62) | 0.52 | 0.75 (0.39, 1.41) | 0.37 |
| | Oropharyngeal cancer | Pulit et al. GWAS/GAME-ON | 2641 | 272 | 0.89 (0.68, 1.15) | 0.36 | 0.75 (0.50, 1.13) | 0.17 | 0.53 (0.27, 1.03) | 0.06 | 0.56 (0.29, 1.07) | 0.08 |
| WC | Oral and oropharyngeal cancer combined | | 6034 | 43 | 0.73 (0.52, 1.02) | 0.07 | 0.64 (0.40, 1.05) | 0.08 | 0.67 (0.36, 1.26) | 0.22 | 0.43 (0.17, 1.08) | 0.08 |
| | Oral cancer | | 2990 | 43 | 0.82 (0.53, 1.26) | 0.36 | 0.66 (0.36, 1.21) | 0.18 | 0.67 (0.32, 1.39) | 0.29 | 0.54 (0.17, 1.76) | 0.31 |
| | Oropharyngeal cancer | Shungin et al. GWAS/GAME-ON | 2641 | 43 | 0.66 (0.43, 1.01) | 0.06 | 0.56 (0.30, 1.05) | 0.07 | 0.37 (0.17, 0.83) | 0.02 | 0.30 (0.09, 0.98) | 0.05 |
| WHR | Oral and oropharyngeal cancer combined | | 6034 | 176 | 0.98 (0.74, 1.29) | 0.88 | 0.98 (0.64, 1.49) | 0.92 | 0.95 (0.45, 2.00) | 0.89 | 1.80 (0.87, 3.71) | 0.11 |
| | Oral cancer | | 2990 | 176 | 1.18 (0.84, 1.65) | 0.35 | 1.00 (0.58, 1.73) | 0.99 | 0.69 (0.29, 1.67) | 0.41 | 2.49 (1.02, 6.12) | 0.05 |
| | Oropharyngeal cancer | Pulit et al. GWAS/GAME-ON | 2641 | 176 | 0.83 (0.59, 1.14) | 0.25 | 0.88 (0.51, 1.50) | 0.63 | 0.93 (0.37, 2.30) | 0.87 | 1.19 (0.50, 2.86) | 0.70 |
| T2D | Oral and oropharyngeal cancer combined | | 6034 | 254 | 0.92 (0.84, 1.01) | 0.09 | 0.85 (0.74, 0.97) | 0.02 | 0.84 (0.71, 1.01) | 0.06 | 0.91 (0.77, 1.09) | 0.31 |
| | Oral cancer | | 2990 | 254 | 0.94 (0.84, 1.05) | 0.27 | 0.84 (0.72, 0.99) | 0.04 | 0.82 (0.66, 1.02) | 0.08 | 0.88 (0.71, 1.08) | 0.22 |
| | Oropharyngeal cancer | Vujkovic et al. GWAS/GAME-ON | 2641 | 254 | 0.94 (0.84, 1.05) | 0.27 | 0.89 (0.73, 1.10) | 0.29 | 1.02 (0.80, 1.30) | 0.88 | 1.00 (0.81, 1.24) | 0.99 |
| $HbA_{1c}$ | Oral and oropharyngeal cancer combined | | 6034 | 37 | 0.56 (0.32, 1.00) | 0.05 | 0.52 (0.23, 1.20) | 0.12 | 0.54 (0.24, 1.21) | 0.14 | 0.37 (0.13, 1.05) | 0.07 |
| | Oral cancer | Wheeler et al. GWAS/GAME-ON (*Lesseur et al., 2016*) | 2990 | 37 | 0.48 (0.24, 0.93) | 0.03 | 0.51 (0.18, 1.41) | 0.19 | 0.44 (0.15, 1.29) | 0.14 | 0.30 (0.09, 1.03) | 0.06 |
| | Oropharyngeal cancer | | 2641 | 37 | 0.66 (0.31, 1.40) | 0.28 | 0.49 (0.15, 1.57) | 0.23 | 0.57 (0.18, 1.85) | 0.35 | 0.43 (0.11, 1.68) | 0.23 |
| FG | Oral and oropharyngeal cancer combined | | 6034 | 28 | 1.06 (0.68, 1.66) | 0.79 | 1.20 (0.62, 2.30) | 0.59 | 1.13 (0.60, 2.12) | 0.71 | 1.11 (0.48, 2.56) | 0.80 |
| | Oral cancer | Lagou et al. GWAS/GAME-ON (*Lesseur et al., 2016*) | 2990 | 28 | 1.05 (0.58, 1.92) | 0.87 | 1.15 (0.48, 2.72) | 0.75 | 0.99 (0.44, 2.23) | 0.99 | 1.25 (0.39, 4.01) | 0.70 |
| | Oropharyngeal cancer | | 2641 | 28 | 1.39 (0.77, 2.51) | 0.28 | 1.24 (0.51, 3.03) | 0.63 | 1.36 (0.59, 3.18) | 0.48 | 1.38 (0.45, 4.18) | 0.58 |
| FI | Oral and oropharyngeal cancer combined | | 6034 | 17 | 0.81 (0.23, 2.89) | 0.75 | 0.75 (0.20, 2.87) | 0.68 | 0.60 (0.03, 10.79) | 0.74 | 0.11 (0.001, 22.47) | 0.43 |
| | Oral cancer | | 2990 | 17 | 0.96 (0.22, 4.16) | 0.96 | 0.46 (0.08, 2.47) | 0.37 | 0.45 (0.01, 19.02) | 0.68 | 0.21 (0.0004, 107.21) | 0.63 |
| | Oropharyngeal cancer | Lagou et al. GWAS/GAME-ON | 2641 | 17 | 0.68 (0.16, 2.87) | 0.59 | 0.66 (0.12, 3.67) | 0.63 | 0.48 (0.05, 4.99) | 0.55 | 0.09 (0.0002, 40.04) | 0.45 |
| SBP | Oral and oropharyngeal cancer combined | *Evangelou et al., 2018* GWAS/GAME-ON (*Lesseur et al., 2016*) | 6034 | 83 | 1.00 (0.97, 1.03) | 0.89 | 0.99 (0.94, 1.03) | 0.55 | 0.98 (0.88, 1.09) | 0.66 | 1.06 (0.92, 1.23) | 0.39 |
| | Oral cancer | | 2990 | 83 | 1.01 (0.96, 1.06) | 0.74 | 0.99 (0.93, 1.04) | 0.65 | 0.95 (0.84, 1.08) | 0.48 | 1.09 (0.90, 1.33) | 0.37 |
| | Oropharyngeal cancer | | 2641 | 83 | 0.99 (0.95, 1.03) | 0.65 | 0.99 (0.94, 1.05) | 0.77 | 1.00 (0.88, 1.13) | 0.94 | 1.03 (0.87, 1.23) | 0.71 |

*Table 1 continued on next page*

*Table 1 continued*

| | | | | | IVW | | Weighted median | | Weighted mode | | MR-Egger | |
|---|---|---|---|---|---|---|---|---|---|---|---|---|
| | Oral and oropharyngeal cancer combined | | 6034 | 64 | 0.93 (0.87, 1.00) | 0.05 | 0.94 (0.86, 1.04) | 0.22 | 1.10 (0.88, 1.38) | 0.42 | 0.99 (0.80, 1.24) | 0.95 |
| | Oral cancer | *Evangelou et al., 2018* GWAS/GAME-ON | 2990 | 64 | 0.95 (0.87, 1.04) | 0.26 | 0.96 (0.86, 1.07) | 0.45 | 1.17 (0.88, 1.56) | 0.28 | 0.97 (0.74, 1.27) | 0.81 |
| DBP | Oropharyngeal cancer | | 2641 | 64 | 0.92 (0.84, 1.00) | 0.05 | 0.94 (0.84, 1.05) | 0.29 | 1.10 (0.86, 1.41) | 0.45 | 1.00 (0.75, 1.30) | 0.93 |

OR are expressed per 1 standard deviation (SD) increase in genetically predicted BMI (4.81 kg/m2), WC (0.09 unit), WHR (0.10 unit), T2D (1-log unit higher odds of T2D), FG (1-log unit increase in mmol/L fasting glucose), FI (1-log unit increase in mmol/L fasting insulin), HbA1c (1-log unit % higher glycated haemoglobin), SBP (1 unit mmHg increase), and DBP (1 unit mmHg increase).

IVW = inverse variance weighted. OR = odds ratio. CI = confidence intervals. p = p-value. BMI = body mass index. WC = waist circumference. WHR = waist–hip ratio. T2D = type 2 diabetes mellitus. FG = fasting glucose. FI = fasting insulin. HbA$_{1c}$ = glycated haemoglobin. SBP = systolic blood pressure. DBP = diastolic blood pressure.

*Figure 2*, *Figure 2—figure supplement 6*). Conversely, there was limited evidence of an effect for FG (OR IVW = 1.06, 95% CI 0.68–1.66, p = 0.79, per 1-log unit increase in mmol/l fasting glucose) (*Table 1*, *Figure 2*, *Figure 2—figure supplement 7*) or FI (OR IVW = 0.81, 95% CI 0.23–2.89, p = 0.75, per 1-log unit increase in mmol/l FI) on combined oral and oropharyngeal cancer risk (*Table 1*, *Figure 2*, *Figure 2—figure supplement 8*).

## Estimated effect of increased blood pressure oral and oropharyngeal cancer risk

Finally, there was limited evidence for an effect of SBP on risk of combined oral and oropharyngeal cancer (OR IVW = 1.00, 95% CI 0.97–1.03, p = 0.89, per 1 unit mmHg increase in systolic blood pressure) (*Table 1*, *Figure 2*, *Figure 2—figure supplement 9*), which did not change when stratified by subsite. However, there was some weak evidence for a protective effect of DBP on risk of combined oral and oropharyngeal cancer (OR IVW = 0.93, 95% CI 0.87–1.00, p = 0.05, per 1 unit mmHg increase in DBP) (*Table 1*, *Figure 2*, *Figure 2—figure supplement 10*).

## Sensitivity analyses

We conducted MR-Egger, weighted median, and weighted mode analyses in addition to IVW (*Table 1*, *Figure 2*). The results of these analyses generally followed the same pattern as the IVW results reported above, however, there were a number of exceptions. The results for HbA$_{1c}$ were not robust to sensitivity testing (p > 0.05 across methods) (*Table 1*, *Figure 2*). In the analysis of T2D on combined oral and oropharyngeal cancer, the weighted median result provided evidence for a weak protective effect (OR weighted median 0.85, 95% CI 0.74–0.97, p = 0.02). This effect appeared mainly in the oral subsite (OR weighted median 0.84, 95% CI 0.72–0.99, p = 0.04). Furthermore, in the analysis of WC on oropharyngeal cancer risk, the weighted mode supported IVW result, providing evidence of a protective effect (OR weighted mode 0.37, 95% CI 0.17–0.83, p = 0.02) (*Table 1*, *Figure 2*).

There was clear evidence of heterogeneity in the SNP effect estimates OR IVW and MR-Egger regression for WHR (Q IVW = 213.04, p = 0.03; Q MR-Egger = 209.24, p = 0.04), T2D (Q IVW = 328.24, p < 0.01; Q MR-Egger = 328.21, p < 0.01), FI (Q IVW = 32.87, p < 0.01; Q MR-Egger = 31.63, p < 0.01), and DBP (Q IVW = 95.82, p < 0.01; Q MR-Egger = 95.22, p < 0.01) (*Supplementary file 2*, Table 2B). MR-Egger intercepts were not strongly indicative of directional pleiotropy (*Supplementary file 2*, Table 2C), but there were outliers present on visual inspection of scatter plots (*Figure 2—figure supplements 11–19*). MR-PRESSO identified 19 outliers for BMI, 2 outliers for WC, 12 outliers for WHR, 23 outliers for T2D, 4 outliers for HbA$_{1c}$, 1 outlier for FG, 3 outliers for FI, 5 outliers for SBP, and 7 outliers for DBP (*Supplementary file 2*, Table 2D–E). When correcting for these outliers, this yielded effects consistent with the primary IVW analysis except for adiposity and T2D instruments, which demonstrated a protective effect on combined oral and oropharyngeal cancer risk when outliers were excluded: BMI (OR IVW = 0.77, 95% CI 0.62–0.94, p = 0.01, per 1 SD in BMI [4.81 kg/m$^2$]); WC (OR IVW = 0.65, 95% CI 0.47–0.89, p = 0.01, per 1 SD in WC [0.09 unit]), and T2D (OR IVW = 0.91, 0.84–0.99, p = 0.03, per 1-log unit higher odds of T2D) (*Supplementary file 2*, Table 2F). Where there was evidence of violation of the negligible measurement error (NOME) assumption for WC, FI, SBP, and DBP (i.e., $I^2$ statistic <0.90) (*Supplementary file 2*, Table 2G), MR-Egger was performed with SIMEX

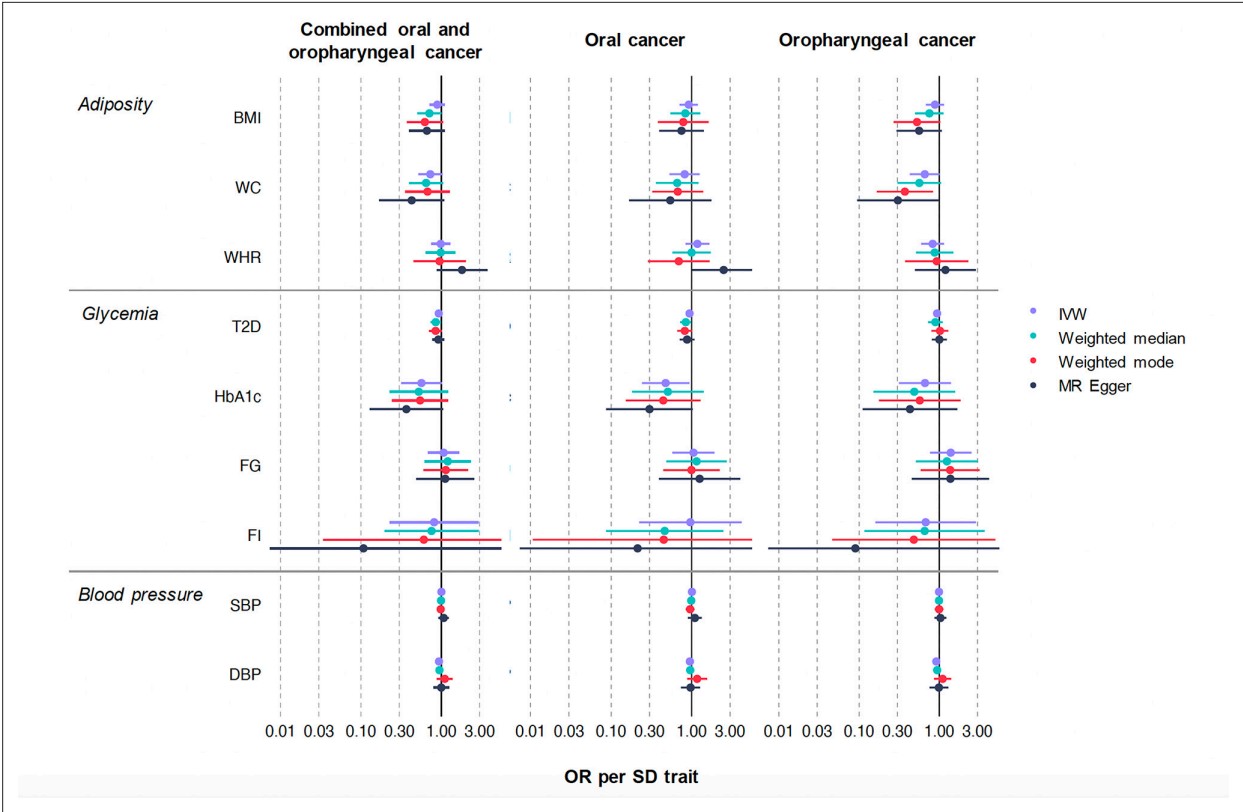

**Figure 2.** Mendelian randomization results of genetically proxied metabolic disorders with risk of oral and oropharyngeal cancer including sensitivity analyses in GAME-ON. Oral and oropharyngeal cancer combined n = 6034, oral cancer n = 2990 and oropharyngeal cancer n = 2641. Abbreviations: IVW, inverse variance weighted; OR, odds ratio with 95% confidence intervals; BMI, body mass index; WC, waist circumference; WHR, waist–hip ratio; T2D, type 2 diabetes mellitus; FG, fasting glucose; FI, fasting insulin; HbA1c, glycated haemoglobin; SBP, systolic blood pressure; DBP, diastolic blood pressure. OR are expressed per 1 standard deviation (SD) increase in genetically predicted BMI (4.81 kg/m²), WC (0.09 unit), WHR (0.10 unit), T2D (1-log unit higher odds of T2D), FG (1-log unit increase in mmol/l fasting glucose), FI (1-log unit increase in mmol/L fasting insulin), HbA1c (1-log unit % higher glycated haemoglobin), SBP (1 unit mmHg increase), and DBP (1 unit mmHg increase).

The online version of this article includes the following figure supplement(s) for figure 2:

**Figure supplement 1.** Power calculations for oral and oropharyngeal analyses in GAME-ON.

**Figure supplement 2.** Forest plots showing Mendelian randomization results for genetically proxied body mass index (BMI) with risk of combined oral and oropharyngeal cancer in GAME-ON.

**Figure supplement 3.** Forest plots showing Mendelian randomization results for genetically proxied waist–hip ratio (WHR) with risk of combined oral and oropharyngeal cancer in GAME-ON.

**Figure supplement 4.** Forest plots showing Mendelian randomization results for genetically proxied waist circumference (WC) with risk of combined oral and oropharyngeal cancer in GAME-ON.

**Figure supplement 5.** Forest plots showing Mendelian randomization results for genetically proxied type 2 diabetes mellitus (T2D) with risk of combined oral and oropharyngeal cancer in GAME-ON.

**Figure supplement 6.** Forest plots showing Mendelian randomization results for genetically proxied glycated haemoglobin (HbA1c) with risk of combined oral and oropharyngeal cancer in GAME-ON.

**Figure supplement 7.** Forest plots showing Mendelian randomization results for genetically proxied fasting glucose (FG) with risk of combined oral and oropharyngeal cancer in GAME-ON.

**Figure supplement 8.** Forest plots showing Mendelian randomization results for genetically proxied fasting insulin (FI) with risk of combined oral and oropharyngeal cancer in GAME-ON.

**Figure supplement 9.** Forest plots showing Mendelian randomization results for genetically proxied systolic blood pressure (SBP) with risk of combined oral and oropharyngeal cancer in GAME-ON.

**Figure supplement 10.** Forest plots showing Mendelian randomization results for genetically proxied diastolic blood pressure (DBP) with risk of combined oral and oropharyngeal cancer in GAME-ON.

**Figure supplement 11.** Scatter plot for body mass index (BMI) with risk of combined oral and oropharyngeal cancer in GAME-ON.

*Figure 2 continued on next page*

correction. SIMEX effects were consistent with the null, except for SBP where an increased risk effect on combined oral and oropharyngeal cancer was found (OR IVW = 1.15, 95% CI 1.05–1.26, p < 0.01, per 1 unit mmHg increase in diastolic blood pressure) (*Supplementary file 2*, Table 2H).

## Evaluating instrument-risk factor effects

Where there was evidence for an effect of BMI, WC, WHR, T2D, HbA$_{1c}$, and DBP on oral and oropharyngeal cancer, we carried out further MR analysis to determine causal effects of these metabolic instruments on established risk HNC risk factors. Adiposity measures showed a strong causal effect on the risk of smoking initiation: BMI [Beta IVW 0.21 (standard error (SE) 0.03), p < 0.001, per 1 SD increase in BMI (4.81 kg/m$^2$)], WC [Beta IVW 0.21 (SE 0.05), p < 0.001, per 1 SD increase in WC (0.09 unit)], and WHR [Beta IVW 0.18 (SE 0.03), p < 0.001, per 1 SD increase in WHR (0.10 unit)] (*Supplementary file 2*, Table 2I). Smaller, yet similar effects were found between adiposity measures and the comprehensive smoking index: BMI [Beta IVW 0.10 (SE 0.01), p < 0.001, per 1 SD increase in BMI (4.81 kg/m$^2$)], WC [Beta IVW 0.10 (SE 0.02), p < 0.001, per 1 SD increase in WC (0.09 unit)], and WHR [Beta IVW 0.09 (SE 0.01), p < 0.001, per 1 SD increase in WHR (0.10 unit)].

There was weaker evidence for an effect of BMI, WC, and genetic liability to T2D on consumption of alcoholic drinks per week: BMI [Beta IVW −0.04 (SE 0.01), p < 0.01, per 1 SD increase in BMI (4.81 kg/m$^2$)], WC [Beta IVW −0.09 (SE 0.02), p < 0.001, per 1 SD increase in WC (0.09 unit)] and T2D [Beta IVW −0.02 (SE 0.01), p < 0.001, per 1-log unit higher odds of T2D]. BMI [Beta IVW 0.04 (SE 0.01), p < 0.001, per 1 SD increase in BMI (4.81 kg/m$^2$)] and WHR [Beta IVW 0.04 (SE 0.02), p = 0.02, per 1 SD increase in WHR (0.10 unit)] were also estimated to increase general risk tolerance. Similarly, increased BMI or WHR and genetic liability to T2D were estimated to decrease educational attainment (years of schooling): BMI [Beta IVW −0.16 (SE 0.02), p < 0.001, per 1 SD increase in BMI (4.81 kg/m$^2$)], WHR [Beta IVW −0.11 (SE 0.02), p < 0.001, per 1 SD increase in WHR (0.10 unit)], and T2D [Beta IVW −0.02 (SE 0.01), p < 0.01, per 1-log unit higher odds of T2D]. However, there was strong evidence of both heterogeneity (*Supplementary file 2*, Table 2J) and genetic pleiotropy (*Supplementary file 2*, Table 2K) across most instrument-risk factor effects. With the exception of alcohol drinks per week, the estimated instrument-risk factor effects remained unchanged following the removal of outlier SNPs detected by MR-PRESSO (*Supplementary file 2*, Table 2L).

## Discussion

In this MR study, we found limited evidence to support a causal role of genetically predicted metabolic traits in oral and oropharyngeal cancer, suggesting the risk may have been previously overestimated in observational studies. However, small effects cannot be excluded given the lack of power to detect them in currently available HNC GWAS. Where weak evidence for an effect was found (i.e., a protective effect of HbA$_{1c}$), these results were not robust to sensitivity analysis, including outlier correction. There was also evidence for instrument-risk factor effects, suggesting smoking may be a mediator between adiposity and HNC.

There are several biological mechanisms linking metabolic traits and cancer, but these have not been well explored in HNC (*Gatenby and Gillies, 2004*; *Grimberg, 2003*; *Tseng et al., 2014*). Dysregulated metabolism is likely linked to the probability a cancer develops and progresses, given that tumours must adapt to satisfy the bioenergetic and biosynthetic demands of chronic cell proliferation via metabolic reprogramming, enhancing or suppressing the activity of metabolic pathways relative to

that in benign tissue (*DeNicola and Cantley, 2015*). In the largest pooled analysis of 17 case–control studies, increasing BMI was associated with a higher risk of overall HNC, but when stratified by subsite the effect was mainly in the larynx (HR 1.42, 95% CI 1.19–1.70 per 5 kg/m$^2$, p < 0.001) (*Gaudet et al., 2015*). Laryngeal cancer was not included in our study given that GWAS summary data were not available for this subsite and future analysis of this region is therefore warranted given this is the most smoking determined cancer. BMI effects on both the oral (HR 1.10, 95% CI 0.97–1.25, p = 0.14) and oropharyngeal cancer (HR 0.98, 95% CI 0.84–1.14, p = 0.77) subsites were consistent with the effects found in our study (oral cancer OR 0.92, 95% CI 0.71–1.19, p = 0.53; oropharyngeal cancer OR 0.89, 0.68–1.15, p = 0.36) (*Gaudet et al., 2015*). Conversely, the same pooled analysis found an increased risk for both WC (HR 1.09, 95% CI 1.03–1.16, p = 0.006) and WHR (HR 1.17, 95% CI 1.02–1.34, p = 0.02), mainly in the oral subsite which were not replicated in our MR analysis. Varying patterns of results for these anthropometric measures have been found when stratifying by smoking status within observational studies (*Gaudet et al., 2015*). The relationship between obesity and HNC is complex. There appears to be a positive association between low BMI (<18.5 kg/m$^2$) and HNC risk, and a protective effect of BMI on HNC risk in current smokers but conversely, a higher risk in never smokers (*Gaudet et al., 2015*). This suggests smoking is a confounder, both as an established risk factor for HNC and in its correlation with weight, with nicotine affecting metabolic energy expenditure, leading to reduced calorie absorption and appetite suppression (*Williamson et al., 1991*). Instrument-risk factor effect estimates from this study suggest smoking is also a mediator, through which metabolic traits such as BMI influence HNC risk. Smoking could be acting as both a mediator and a confounder, since the relationship between BMI and smoking is bi-directional (i.e., smoking reduces BMI and higher BMI in turn increases the likelihood of smoking), which has been demonstrated in previous MR studies (*Carreras-Torres et al., 2018*; *Taylor et al., 2019*).

Despite metabolic syndrome (including hypertension, central obesity, elevated triglyceride, low High-density lipoprotein cholesterol (HDL-C), and insulin resistance) being strongly associated with common cancers such as colorectal and breast (*Esposito et al., 2012*), this does not appear to be the case in HNC. A recent prospective study of 474,929 participants from UK Biobank investigating the effect of metabolic syndrome suggested those with the condition had no increased HNC risk (HR 1.05, 95% CI 0.90–1.22, p = 0.560) (*Jiang et al., 2021*). No definitive causal effects were detected for individual components of metabolic syndrome components either, supporting our MR results. While another large meta-analysis found individuals with T2D have an elevated risk of oral cancer (*Gong et al., 2015*), other more recent studies have found this effect to be mostly in laryngeal subsite (HR 1.25, 95% CI 1.12–1.40) which again we could not investigate in this study (*Kim et al., 2021*). Hypertension is the most consistently reported metabolic trait to have an observational association with HNC risk across the subsites (*Christakoudi et al., 2020*; *Kim et al., 2021*; *Kim et al., 2019*; *Seo et al., 2020*; *Stocks et al., 2012*). We did not identify a clear effect of either SBP or DBP on oral or oropharyngeal cancer using MR, again suggesting the possibility of residual confounding in observational studies.

MR was employed in this study in an attempt to overcome the drawbacks of conventional epidemiological studies. However, there are a number of limitations with using this approach and if MR assumptions are violated, this too can generate spurious conclusions. While there was no evidence of weak instrument bias (F statistics >10), there was heterogeneity present in at least four of the instruments (WHR, T2D, FI, and DBP). This is expected to some extent, given that we are instrumenting multiple biological pathways that contribute to complex metabolic phenotypes. The use of multiple related instruments for each metabolic trait may, however, provide some additional confidence in the overall findings. Given the low percentage of variation explained ($R^2$) for some instruments, as well as the relatively small number of oral and oropharyngeal cancer cases, power to detect an effect may have been an issue in some of our analyses.

As with observational studies, there may be issues of measurement error or misclassification in genetic epidemiology, given BMI is simply a function of mass and height and does not specifically measure adiposity. However, BMI has been shown to be an acceptable proxy when used in large samples sizes, correlating with both total body fat (*Browning et al., 2011*) and total abdominal adipose tissue (*Ross et al., 1992*), which is thought to present a greater health risk than fat deposited elsewhere. Furthermore, we used a range of adiposity measures including WC and WHR, which may be better proxies of abdominal adiposity, compared to BMI (*Lee et al., 2008*).

Risk tolerance is challenging to instrument genetically due to measurement error (e.g., as a result of reporting bias) and because it is socially patterned, time-varying as well as context and culture-dependent (*Gormley et al., 2022*). It may also be a poor proxy for sexual behaviour, despite genetic correlation with these phenotypes given that pleiotropy with other traits such as smoking may be present (*Mills et al., 2021*). However, genetic instruments are not available specifically for oral sex, which is the conceptually relevant exposure and likely mode of HPV transmission.

SNPs used to proxy these metabolic traits, particularly adiposity measures BMI, WC, and WHR were also strongly associated with smoking. Repeating this analysis in an updated, better powered GWAS is required in order to exclude any potential small effects of metabolic traits on HNC risk via smoking. Given the heterogeneity of these complex metabolic traits, future work could further examine their pathway-specific effects (*Udler et al., 2018*).

Overall, there was limited evidence for an effect of genetically proxied metabolic traits on oral and oropharyngeal cancer risk. These findings suggest metabolic traits may not be effective modifiable risk factors to prioritize as part of future prevention strategies in HNC, however, small effects cannot be excluded and further replication in larger GWAS is needed. The effect of metabolic traits on the risk of this disease may have been overestimated in previous observational studies, but these cannot be directly compared given the differences in methodological approaches and the interpretation of estimates. Smoking appears to act as a mediator in the relationship between obesity and HNC. Although there is no clear evidence that changing body mass will reduce or increase the risk of HNC directly, dental and medical teams should be aware of the risk of smoking in those who are overweight and therefore the greater risk of cancer when providing smoking cessation and appropriate weight loss advice.

## Materials and methods

Two-sample MR was performed using published summary-level data from the largest available GWAS for each metabolic trait.

### Exposure summary statistics for metabolic traits

To instrument metabolic traits, we selected genetic variants associated ($p < 5 \times 10^{-8}$) with traits of interest identified by previously conducted GWAS (*Supplementary file 1*). Clumping was performed in the TwoSampleMR package to ensure SNPs in each instrument were independent ($r^2 < 0.001$). This accounted for any potential linkage disequilibrium between SNPs, which can lead to overestimation of instrument strength and overly precise effect estimates. Following clumping, genetic instruments were comprised of: 312 SNPs for BMI, from a GWAS meta-analysis of 806,834 individuals of European ancestry, including the Genetic Investigation of ANthropometric Traits (GIANT) consortium and UK Biobank (*Pulit et al., 2019*) and 209 SNPs for WHR extracted from the same GWAS in 697,734 individuals (*Pulit et al., 2019*). Forty-five SNPs for WC were taken from a GWAS meta-analysis describing 224,459 individuals of mainly European ancestry (*Shungin et al., 2015*), 275 SNPs for T2D from the DIAMANTE (DIAbetes Meta-ANalysis of Trans-Ethnic association studies) consortium of 228,499 cases and 1,178,783 controls (*Vujkovic et al., 2020*), 33 SNPs for FG and 18 SNPs for FI, obtained from a GWAS published by the MAGIC (Meta-Analyses of Glucose and Insulin-Related Traits) Consortium ($N = 151,188$ and 105,056 individuals of European descent, respectively) (*Lagou et al., 2021*); 58 SNPs for HbA$_{1c}$, taken from a meta-analysis of 88,355 individuals from European cohorts (*Wheeler et al., 2017*); finally, 105 and 78 SNPs for SBP and DBP, respectively, were extracted from a GWAS meta-analysis of over 1million participants in UK Biobank and the International Consortium of Blood Pressure Genome Wide Association Studies (ICBP) (*Evangelou et al., 2018*; *Supplementary file 1*).

### Outcome summary statistics for oral and oropharyngeal cancer

We estimated the effects of metabolic traits on risk of oral and oropharyngeal cancer by extracting exposure SNPs (*Supplementary file 1*) from the largest available GWAS performed on 6034 cases and 6585 controls from 12 studies which were part of the Genetic Associations and Mechanisms in Oncology (GAME-ON) Network (*Lesseur et al., 2016*). Full details of the included studies, as well as the genotyping and imputation performed, have been described previously (*Dudding et al., 2018*; *Lesseur et al., 2016*). In brief, the study population included participants from Europe (45.3%), North

America (43.9%), and South America (10.8%). Cancer cases comprised the following the International Classification of Diseases (ICD-10) codes: oral (C02.0–C02.9, C03.0–C03.9, C04.0–C04.9, C05.0–C06.9), oropharyngeal (C01.9, C02.4, C09.0–C10.9), hypopharyngeal (C13.0–C13.9), overlapping (C14 and combination of other sites), and 25 cases with unknown code (other). A total of 954 individuals with cancers of hypopharynx, unknown code or overlapping cancers were excluded. Genomic DNA isolated from blood or buccal cells was genotyped at the Center for Inherited Disease Research (CIDR) using an Illumina OncoArray, custom designed for cancer studies by the OncoArray Consortium (*Consortium, 2013*). Principle components analysis was performed using approximately 10,000 common markers in low linkage disequilibrium (LD) ($r^2 < 0.004$), minor allele frequency >0.05 and 139 population outliers were removed.

Given the differential association of potential risk factors at each subsite (i.e., smoking, alcohol and HPV infection) (*Thomas et al., 2018*), we performed stratified MR analyses for oral and oropharyngeal cancer to evaluate potential heterogeneity in effects. For this, we used GWAS summary data on a subset of 2990 oral and 2641 oropharyngeal cases and the 6585 common controls in the GAME-ON GWAS (*Lesseur et al., 2016*).

## Statistical analysis

Two-sample MR was conducted using the 'TwoSampleMR' package in R (version 3.5.3), by integrating SNP associations for each metabolic trait (exposure, sample 1) with those for oral and oropharyngeal cancer in GAME-ON (outcome, sample 2). For exposures, we only used genetic variants reaching GWAS significance ($p < 5 \times 10^{-8}$). The nearest gene was identified using SNPsnap and a distance of ±500 kb (*Pers et al., 2015*). Firstly, metabolic trait-associated SNPs were extracted from oral and oropharyngeal cancer summary statistics. Exposure and outcome summary statistics were harmonized using the '*harmonise_data*' function of the TwoSampleMR package so that variant effect estimates corresponded to the same allele. Palindromic SNPs were identified and corrected using allele frequencies where possible (alleles were aligned when minor allele frequencies were <0.3, or were otherwise excluded). For each SNP in each exposure, individual MR effect estimates were calculated using the Wald method (SNP-outcome beta/SNP-exposure beta) (*Wald, 1940*). Multiple SNPs were then combined into multi-allelic instruments using random-effects IVW meta-analysis.

IVW estimates may be vulnerable to bias if genetic instruments are invalid and are only unbiased in the absence of horizontal pleiotropy or when horizontal pleiotropy is balanced (*Hemani et al., 2018*). We therefore performed additional sensitivity analyses to evaluate the potential for unbalanced horizontal pleiotropy using weighted median (*Bowden et al., 2016a*), weighted mode (*Hartwig et al., 2017*), and MR-Egger (*Bowden et al., 2015*) methods which are described in detail elsewhere (*Lawlor et al., 2019*). In short, the weighted median stipulates that at least 50% of the weight in the analysis stems from valid instruments. Weighted mode returns an unbiased estimate of the causal effect if the cluster with the largest weighted number of SNPs for the weighted model are all valid instruments. Instruments are weighted by the inverse variance of the SNP-outcome association (*Hartwig et al., 2017*).

Finally, MR-Egger provides reliable effect estimates even if variants are invalid and the Instrument Strength Independent of Direct Effect (InSIDE) assumption is violated (*Bowden et al., 2015*). The InSIDE assumption states that the association between genetic instrument and exposure should not be correlated with an independent path from instrument to the outcome. In the presence of unbalanced pleiotropy when the InSIDE assumption is violated, then the MR-Egger result may be biased (*Lawlor et al., 2019*). Gene variants must be valid instruments and where there was evidence of violation of the NOME assumption (*Bowden et al., 2016b*), this was assessed using the $I^2$ statistic and MR-Egger was performed with simulation extrapolation (SIMEX) correction for bias adjustment (*Bowden et al., 2016b*). The variance of each trait explained by the genetic instrument ($R^2$) was estimated and used to perform power calculations (*Brion et al., 2013*). F-statistics were also generated. An F-statistic lower than 10 was interpreted as indicative of a weak instrument bias (*Lawlor et al., 2008*). To further assess the robustness of MR estimates, we examined evidence of heterogeneity across individual SNPs using the Cochran Q-statistic, which indicates the presence of invalid instruments (e.g., due to horizontal pleiotropy), if Q is much larger than its degrees of freedom (No. of instrumental variables minus 1) (*Bowden et al., 2018*). MR-PRESSO (Mendelian Randomization Pleiotropy RESidual Sum and Outlier)

was used to detect and correct for potential outliers (where *Q*-statistic p < 0.05) (*Verbanck et al., 2018*).

## Instrument-risk factor effects

Where there was evidence for an effect of a metabolic trait on oral or oropharyngeal cancer risk in the primary MR analysis, we conducted further evaluation of the metabolic instruments onto established HNC risk factors using two-sample MR. The largest available GWAS were used for smoking initiation (a binary phenotype indicating whether an individual had ever smoked in their life versus never smokers) (*n* = 1,232,091) and alcoholic drinks per week (defined as the average number of drinks per week aggregated across all types of alcohol, *n* = 941,280) from the GWAS and Sequencing Consortium of Alcohol and Nicotine use (GSCAN) study (*Liu et al., 2019*). The comprehensive smoking index, a quantitative lifetime measure of smoking behaviour derived from 462,690 individuals from UK Biobank was also employed. A 1 standard deviation (SD) increase in the index is equivalent to an individual smoking 20 cigarettes a day for 15 years and stopping 17 years ago, or an individual smoking 60 cigarettes a day for 13 years and stopping 22 years ago.

Summary statistics were also obtained from a GWAS of general risk tolerance (*n* = 939,908), derived from a meta-analysis of UK Biobank (*n* = 431,126) binary question '*Would you describe yourself as someone who takes risks?*' and the 23andMe (*n* = 508,782) question '*Overall, do you feel comfortable or uncomfortable taking risks?*'. The GWAS of risk tolerance was based on one's tendency or willingness to take risks, making them more likely to engage in risk-taking behaviours more generally (*Karlsson Linnér et al., 2019*). A strong genetic correlation between sexual behaviours and risk tolerance has been shown previously (*Gormley et al., 2022*). Finally, given the known association between HNC and lower socioeconomic position, we used MR to examine educational attainment (defined by years of schooling) (*Lee et al., 2018*). Outcome beta estimates reflect the standard deviation of the phenotype.

## Acknowledgements

M.G. was a National Institute for Health Research (NIHR) academic clinical fellow and is currently supported by a Wellcome Trust GW4-Clinical Academic Training PhD Fellowship. This research was funded in part, by the Wellcome Trust [Grant number 220530/Z/20/Z]. For the purpose of open access, the author has applied a CC BY public copyright licence to any Author Accepted Manuscript version arising from this submission. R.C.R. is a de Pass VC research fellow at the University of Bristol. J.T. is supported by an Academy of Medical Sciences (AMS) Springboard award, which is supported by the AMS, the Wellcome Trust, Global Challenges Research Fund (GCRF), the Government Department of Business, Energy and Industrial strategy, the British Heart Foundation and Diabetes UK (SBF004\1079). A.R.N. was supported by the National Institute for Health Research (NIHR) Bristol Biomedical Research Centre which is funded by the National Institute for Health Research (NIHR) and is a partnership between University Hospitals Bristol NHS Foundation Trust and the University of Bristol. Department of Health and Social Care disclaimer: The views expressed are those of the authors and not necessarily those of the NHS, the NIHR or the Department of Health and Social Care. This publication presents data from the Head and Neck 5000 which contributes to international VOYAGER and HEADSpAcE head and neck cancer consortia. The Head and Neck 5000 study was a component of independent research funded by the National Institute for Health Research (NIHR) under its Programme Grants for Applied Research scheme (RP-PG-0707-10034). The views expressed in this publication are those of the author(s) and not necessarily those of the NHS, the NIHR or the Department of Health. Core funding was also provided through awards from Above and Beyond, University Hospitals Bristol and Weston Research Capability Funding and the NIHR Senior Investigator award to A.R.N. Human papillomavirus (HPV) serology was supported by a Cancer Research UK Programme Grant, the Integrative Cancer Epidemiology Programme (C18281/A20919). The VOYAGER study was supported in part by the US National Institute of Dental and Craniofacial Research (NIDCR; R01 DE025712). The genotyping of the HNC cases and controls was performed at the Center for Inherited Disease Research (CIDR) and funded by the US National Institute of Dental and Craniofacial Research (NIDCR; 1X01HG007780-0). E.E.V., C.B., and D.L. are supported by Diabetes UK (17/0005587). E.E.V. and C.B. are supported by the World Cancer Research Fund (WCRF UK), as part of the World Cancer Research Fund International grant programme (IIG_2019_2009). M.G., T.D., G.D.S., E.E.V., R.C.R., and C.B. are part of the Medical

Research Council Integrative Epidemiology Unit at the University of Bristol supported by the Medical Research Council (MC_UU_00011/1, MC_UU_00011/5, MC_UU_00011/6, MC_UU_00011/7).

## Additional information

### Funding

| Funder | Grant reference number | Author |
|---|---|---|
| Wellcome Trust | 220530/Z/20/Z | Mark Gormley |
| Diabetes UK | SBF004\1079 | Jessica Tyrrell |
| National Institute for Health and Care Research | RP-PG-0707-10034 | Andrew R Ness |
| Cancer Research UK | C18281/A20919 | Andrew R Ness |
| National Institute of Dental and Craniofacial Research | R01 DE025712 and 1X01HG007780-0 | Andrew R Ness |
| Diabetes UK | 17/0005587 | Emma E Vincent |
| World Cancer Research Fund | IIG_2019_2009 | Emma E Vincent |
| Medical Research Council | MC_UU_00011/1 | George Davey Smith |
| Medical Research Council | MC_UU_00011/5 | George Davey Smith |
| Medical Research Council | MC_UU_00011/6 | George Davey Smith |
| Medical Research Council | MC_UU_00011/7 | George Davey Smith |

The funders had no role in study design, data collection, and interpretation, or the decision to submit the work for publication. For the purpose of Open Access, the authors have applied a CC BY public copyright license to any Author Accepted Manuscript version arising from this submission.

### Author contributions

Mark Gormley, Conceptualization, Data curation, Formal analysis, Funding acquisition, Investigation, Visualization, Methodology, Writing – original draft, Project administration, Writing – review and editing; Tom Dudding, Conceptualization, Methodology, Writing – review and editing; Steven J Thomas, Miranda Pring, Danny Legge, Supervision, Writing – review and editing; Jessica Tyrrell, Supervision, Methodology, Writing – review and editing; Andrew R Ness, Conceptualization, Supervision, Writing – review and editing; George Davey Smith, Emma E Vincent, Conceptualization, Supervision, Methodology, Writing – original draft, Writing – review and editing; Rebecca C Richmond, Conceptualization, Supervision, Methodology, Writing – review and editing; Caroline Bull, Conceptualization, Formal analysis, Supervision, Validation, Investigation, Visualization, Methodology, Writing – original draft, Writing – review and editing

### Author ORCIDs

Mark Gormley ⓘ http://orcid.org/0000-0001-5733-6304
Danny Legge ⓘ http://orcid.org/0000-0002-3897-5861
George Davey Smith ⓘ http://orcid.org/0000-0002-1407-8314
Emma E Vincent ⓘ http://orcid.org/0000-0002-8917-7384
Caroline Bull ⓘ http://orcid.org/0000-0002-2176-5120

### Ethics

Publicly available summary-level data only used in this study. Application entitled 'Investigating aetiology, associations and causality in diseases of the head and neck' (Project ID: 40644) covers use of all UK Biobank data in this study and dbGaP application made for accessing OncoArray: Oral and Pharynx Cancer; study accession number: phs001202.v1.p1 data entitled 'Investigating risk factors in head and neck cancer using Mendelian randomization' (Project ID 24266). All studies included as part of the GAME-ON network obtained approval and consent from their respective institutions.

Decision letter and Author response
Decision letter https://doi.org/10.7554/eLife.82674.sa1
Author response https://doi.org/10.7554/eLife.82674.sa2

## Additional files

### Supplementary files

• Supplementary file 1. Genetic variants associated with metabolic traits of interest.

• Supplementary file 2. Showing tables 2A through 2L. Supplementary file 2, Table 2A. Assessing weak instrument bias ($F$-statistic) and proportion of variance in the phenotype ($R^2$) explained by metabolic phenotype instruments. Abbreviations: BMI, body mass index; WC, waist circumference; WHR, waist–hip ratio; T2D, type 2 diabetes mellitus; $HbA_{1c}$, glycated haemoglobin; FG, fasting glucose; FI, fasting insulin; SBP, systolic blood pressure; DBP, diastolic blood pressure. Supplementary file 2, Table 2B. Assessing heterogeneity of single nucleotide polymorphism effect estimates in inverse variance weighted (IVW) and MR-Egger regression for metabolic disorder analysis. Abbreviations: $Q$, $Q$-statistic; df, degrees of freedom; p, p-value; BMI, body mass index; WC, waist circumference; WHR, waist–hip ratio; T2D, type 2 diabetes mellitus; $HbA_{1c}$, glycated haemoglobin; FG, fasting glucose; FI, fasting insulin; SBP, systolic blood pressure; DBP, diastolic blood pressure. Supplementary file 2, Table 2C. Assessing directional pleiotropy through MR-Egger intercept for metabolic disorder analysis. Abbreviations: SE, standard error; p, p-value; BMI, body mass index; WC, waist circumference; WHR, waist–hip ratio; T2D, type 2 diabetes mellitus; $HbA_{1c}$, glycated haemoglobin; FG, fasting glucose; FI, fasting insulin; SBP, systolic blood pressure; DBP, diastolic blood pressure. Supplementary file 2, Table 2D. MR-PRESSO outliers detected results in the analysis of metabolic disorders on combined oral and oropharyngeal cancer risk. Abbreviations: $Q$-stat, Cochran's $Q$-statistic; BMI, body mass index; WC, waist circumference; WHR, waist–hip ratio; T2D, type 2 diabetes mellitus; $HbA_{1c}$, glycated haemoglobin; FG, fasting glucose; FI, fasting insulin; SBP, systolic blood pressure; DBP, diastolic blood pressure. Supplementary file 2, Table 2E. MR-PRESSO results for metabolic disorders on combined oral and oropharyngeal cancer. Abbreviations: SE, standard error; p, p-value; BMI, body mass index; WC, waist circumference; WHR, waist–hip ratio; T2D, type 2 diabetes mellitus; $HbA_{1c}$, glycated haemoglobin; FG, fasting glucose; FI, fasting insulin; SBP, systolic blood pressure; DBP, diastolic blood pressure. Supplementary file 2, Table 2F. Outlier corrected results in the analysis of metabolic disorders on combined oral and oropharyngeal cancer risk. Abbreviations: SE, standard error; OR, odds ratio; CI, confidence intervals; IVW, inverse variance weighted; BMI, body mass index; WC, waist circumference; WHR, waist–hip ratio; T2D, type 2 diabetes mellitus; $HbA_{1c}$, glycated haemoglobin; FG, fasting glucose; FI, fasting insulin; SBP, systolic blood pressure; DBP, diastolic blood pressure. Supplementary file 2, Table 2G. Assessing violation of the NO measurement error (NOME) assumption for instruments used in MR-Egger regression. Abbreviations: $I^2$, I-squared statistic; BMI, body mass index; WC, waist circumference; WHR, waist–hip ratio; T2D, type 2 diabetes mellitus; $HbA_{1c}$, glycated haemoglobin; FG, fasting glucose; FI, fasting insulin; SBP, systolic blood pressure; DBP, diastolic blood pressure. Supplementary file 2, Table 2H. SIMEX correction MR-Egger regression results for where NO measurement error (NOME) assumption may have been violated ($I^2 < 0.90$). Abbreviations: OR, odds ratio; CI, confidence intervals; WC, waist circumference; FI, fasting insulin; SBP, systolic blood pressure; DBP, diastolic blood pressure. Supplementary file 2, Table 2I. Mendelian randomization results evaluating instrument-risk factor effects. Abbreviations: IVW, inverse variance weighted; OR, odds ratio; CI, confidence intervals; p, p-value; BMI, body mass index; WC, waist circumference; WHR, waist–hip ratio; T2D, type 2 diabetes mellitus; $HbA_{1c}$, glycated haemoglobin; DBP, diastolic blood pressure. OR are expressed per 1 standard deviation (SD) increase in genetically predicted BMI ($4.81$ kg/m$^2$), WC ($0.09$ unit), WHR ($0.10$ unit), T2D (1-log unit higher odds of T2D), $HbA_{1c}$ (1-log unit % higher glycated haemoglobin), and DBP (1 unit mmHg increase). Outcome beta estimates reflect the standard deviation of the phenotype. Supplementary file 2, Table 2J. Assessing heterogeneity in Mendelian randomization results evaluating instrument-risk factor effects. Abbreviations: $Q$, $Q$-statistic; df, degrees of freedom; p, p-value; BMI, body mass index; WC, waist circumference; WHR, waist–hip ratio; T2D, type 2 diabetes mellitus; $HbA_{1c}$, glycated haemoglobin; DBP, diastolic blood pressure. Supplementary file 2, Table 2K. Assessing directional pleiotropy in Mendelian randomization results evaluating instrument-risk factor effects. Abbreviations: SE, standard error; p, p-value; BMI, body mass index; WC, waist circumference; WHR, waist–hip ratio; T2D, type 2 diabetes mellitus; $HbA_{1c}$, glycated haemoglobin; DBP, diastolic blood pressure. Supplementary file 2, Table 2L. Outlier corrected Mendelian randomization results evaluating

instrument-risk factor effects. Abbreviations: IVW, inverse variance weighted; OR, odds ratio; CI, confidence intervals; p, p-value; BMI, body mass index; WC, waist circumference; WHR, waist–hip ratio; T2D, type 2 diabetes mellitus; HbA$_{1c}$, glycated haemoglobin; DBP, diastolic blood pressure. OR are expressed per 1 standard deviation (SD) increase in genetically predicted BMI (4.81 kg/m$^2$), WC (0.09 unit), WHR (0.10 unit), T2D (1-log unit higher odds of T2D), HbA$_{1c}$ (1-log unit % higher glycated haemoglobin), and DBP (1 unit mmHg increase). Outcome beta estimates reflect the standard deviation of the phenotype.

• Transparent reporting form

## Data availability

Summary-level analysis was conducted using publicly available GWAS data as cited. Full summary statistics for the GAME-ON outcome data GWAS can be accessed via dbGAP (OncoArray: Oral and Pharynx Cancer; study accession number: phs001202.v1.p1, August 2017) at: https://www.ncbi.nlm.nih.gov/projects/gap/cgi-bin/study.cgi?study_id=phs001202.v1.p1 (*Lesseur et al., 2016*). This data is also available via the IEU OpenGWAS project (https://gwas.mrcieu.ac.uk/). All exposure data used in this study is publicly available from the relevant studies as described below. Data for BMI, WC and WHR GWAS was downloaded from the Genetic Investigation of ANthropometric Traits (GIANT) consortium https://portals.broadinstitute.org/collaboration/giant/index.php/GIANT_consortium_data_files (*Pulit et al., 2019*; *Shungin et al., 2015*) and UK Biobank (http://www.ukbiobank.ac.uk). T2D data was downloaded from the DIAMANTE (DIAbetes Meta-ANalysis of Trans-Ethnic association studies) consortium from: https://kp4cd.org/node/169 (*Vujkovic et al., 2020*). Data for FG, FI, and HbA1c were obtained from GWAS published by the MAGIC (Meta-Analyses of Glucose and Insulin-Related Traits) Consortium, available for download from: https://magicinvestigators.org/downloads/ (*Lagou et al., 2021*),. Finally, data for SBP and DBP were extracted from a GWAS meta-analysis of participants in UK Biobank and UK Biobank (http://www.ukbiobank.ac.uk) and the International Consortium of Blood Pressure Genome Wide Association Studies (ICBP), available via dbGAP (International Consortium for Blood Pressure (ICBP), study accession number: phs000585.v2.p1, October 2016) at https://www.ncbi.nlm.nih.gov/projects/gap/cgi-bin/study.cgi?study_id=phs000585.v2.p1 (*Evangelou et al., 2018*). Instrument-risk factor analysis outcome summary-level data were derived from the GWAS and Sequencing Consortium of Alcohol and Nicotine use (GSCAN) and UK Biobank and UK Biobank (http://www.ukbiobank.ac.uk) for alcoholic drinks per week https://conservancy.umn.edu/handle/11299/201564 (*Liu et al., 2019*) and the comprehensive smoking index (*Wootton et al., 2020*). Data for risk tolerance and educational attainment were taken from Social Science Genetic Association Consortium (SSGAC) data available from http://www.thessgac.org/data (*Karlsson Linnér et al., 2019*; *Lee et al., 2018*). MR analyses were conducted using the 'TwoSampleMR' package in R (version 3.5.3). A copy of the code and all data files used in this study are available at GitHub (https://github.com/MGormley12/metabolic_trait_hnc_mr.git copy archived at *Gormley, 2023*).

The following previously published datasets were used:

| Author(s) | Year | Dataset title | Dataset URL | Database and Identifier |
|---|---|---|---|---|
| Lesseur C | 2017 | OncoArray: Oral and Pharynx Cancer | https://www.ncbi.nlm.nih.gov/projects/gap/cgi-bin/study.cgi?study_id=phs001202.v1.p1 | NCBI BioProject, phs001202.v1.p1 |
| Pulit SL | 2018 | Summary-level data from meta-analysis of fat distribution phenotypes in UK Biobank and GIANT | https://doi.org/10.5281/zenodo.1251813 | Zenodo, 10.5281/zenodo.1251813 |
| Shungin D | 2015 | GWAS Anthropometric 2015 Waist Summary Statistics | https://portals.broadinstitute.org/collaboration/giant/index.php/GIANT_consortium_data_files | Genetic Investigation of ANthropometric Traits (GIANT) consortium, GIANT_consortium_data_files |

*Continued on next page*

*Continued*

| Author(s) | Year | Dataset title | Dataset URL | Database and Identifier |
|---|---|---|---|---|
| Vujkovic M | 2020 | DIAMANTE (European) T2D GWAS | https://kp4cd.org/node/169 | DIAMANTE (DIAbetes Meta-ANalysis of Trans-Ethnic association studies), GWAS_DIAMANTE_eu |
| Evangelou E | 2016 | International Consortium for Blood Pressure (ICBP) | https://www.ncbi.nlm.nih.gov/projects/gap/cgi-bin/study.cgi?study_id=phs000585.v2.p1 | NCBI BioProject, phs000585.v2.p1 |
| Liu M | 2019 | Data Related to Association studies of up to 1.2 million individuals yield new insights into the genetic etiology of tobacco and alcohol use | https://doi.org/10.13020/3b1n-ff32 | DRUM, 10.13020/3b1n-ff32 |

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
