## [Editor Report]

This work presents valuable findings on the causal association of metabolic traits and head and neck cancers. The evidence supporting the conclusion is convincing, with rigorous and comprehensive data analysis. The work will be of interest to cancer epidemiologists, especially those working on head and neck cancer.

---

## [Decision Letter]

**Decision letter after peer review:**

Thank you for submitting your article "Evaluating the effect of metabolic traits on oral and oropharyngeal cancer risk using Mendelian randomization" for consideration by *eLife*. Your article has been reviewed by 2 peer reviewers, and the evaluation has been overseen by a Reviewing Editor and a Senior Editor. The following individual involved in the review of your submission has agreed to reveal their identity: Susan Martin (Reviewer #1).

As is customary in *eLife*, the reviewers have discussed their critiques with one another and with the Reviewing and Senior Editors. The decision was reached by consensus. What follows below is the edited compilation of the essential and ancillary points provided by reviewers in their critiques and in their interaction post-review. Please submit a revised version that addresses these concerns directly. Although we expect that you will address these comments in your response letter, we also need to see the corresponding revision clearly marked in the text of the manuscript. Some of the reviewers' comments may seem to be simple queries or challenges that do not prompt revisions to the text. Please keep in mind, however, that readers may have the same perspective as the reviewers. Therefore, it is essential that you amend or expand the text to clarify the narrative accordingly.

Essential revisions:

Please better explain the reasons for not including FinnGen as an outcome GWAS source. As suggested by the reviewer, 'MR could have been run using the summary statistics from FinnGen, and these MR results were then meta-analysed with those from your original cohort. ' As rightly pointed out by the reviewers, the disease outcomes are available in FinnGen, and using this strategy would increase sample size and statistical power.

Another issue pointed out by the reviewers is the possibility of SNPs for the metabolic traits to act pleiotropically on the other metabolic traits. In case this possibility is true, it is hard to differentiate if those SNPs are acting through the studied trait or through another mechanism. Also, the choice of a threshold of 0.001 for the linkage disequilibrium needs to be better justified, as this option might affect the number of SNPs related to BMI. Moreover, the authors could elaborate on the reasons -for using SNPs identified in from the multi-ancestry HbA1c MAGIC GWAS rather than the European-specific.

Furthermore, the reviewers raised questions about the suitability of measuring lifetime smoking as ever vs never and using risk-taking as a proxy for HPV infection.

*Reviewer #1 (Recommendations for the authors):*

This is a thorough well-conducted MR analysis, just some minor points to be considered:

– Make the aim clearer in the Introduction.

– Was there any reason why FinnGen was not included as an outcome GWAS source? MR could have been run using the summary statistics from FinnGen and these MR results were then meta-analysed with those from your original cohort (which doesn't appear to contain FinnGen or have any risk of overlap). Both disease outcomes exist in FinnGen, and while there are few cases of oropharyngeal cancer, there are many for oral cancer, which would greatly increase your sample size and thus increase power. Unless there is a specific reason not to include FinnGen, I recommend that the analysis should be re-done (at least as a sensitivity analysis) including the publicly available FinnGen summary statistics.

– "A meta-analysis of observational studies investigating T2D with oral and oropharyngeal subsites, showed an increased risk ratio (RR) of 1.15, 95%CI 1.02-1.29, P heterogeneity = 0.277" – did you mean to give only the P-value for the heterogeneity test here? Seems like the P-value relating to the RR should also be included.

– "investigate if an exposure (X), is associated with a trait e.g., body mass index or disease outcome (Y)" – this sentence is misleading to the reader as you have used BMI as an exposure (Figure 1), so this sentence should be re-phrased.

– Your linkage disequilibrium (R2) threshold of 0.001 seems very low – was there any specific reason behind this choice of threshold? Maybe briefly explain the reasoning behind this choice in the text. As a result, you end up with few variants – for example, you only have 312 SNPs for BMI – other studies using BMI as an exposure in MR analyses have used ~700 variants from the GIANT consortium.

– When you describe the MAGIC cohort you do not mention the ancestry, as you did with the other cohorts – please also include this.

– Why did you use SNPs identified in the multi-ancestry HbA1c MAGIC GWAS rather than the European-specific one when this is available?

– Did you use proxy SNPs when your genetic variants of interest were not available in the outcome GWAS summary statistics? If so, how were these proxy SNPs identified in terms of R2 thresholds for LD with the index variant, distance from the index variant, data sources used, etc.?

– "We only used genetic variants reaching GWAS significance (p <5x10-8)." – you should explicitly state that this is only for the exposure, and not the outcome.

– "individual had ever smoked in their life versus never smokers" – is this the best smoking measure? Surely a variable that measures whether someone was a regular smoker or not would be better if available – maybe expand on this in limitations.

– You state that there are known strong genetic correlations between risk-taking and sexual behavior – however, it still may not be a great proxy for the latter – maybe state this in your limitations.

– "SNPs used to proxy these metabolic traits, particularly obesity-related measures BMI, WC, and WHR were strongly associated with smoking." – you mention this very briefly at the end, and expand on this potential limitation.

– "Although there is no clear evidence that changing body mass will reduce of increase the risk" – 'or' instead of 'of'?

*Reviewer #2 (Recommendations for the authors):*

1. Do any of the SNPs for the metabolic traits act pleiotropically on the other metabolic traits? If so, it would be hard to know whether those SNPs are acting through the studied trait or through another mechanism (like another correlated metabolic trait).

2. Is there any way to replicate these findings?

3. It is unclear how the last sentence of the abstract can be used as a conclusion based on the results portion of the abstract. Suggest revising.

4. Suggest clearly using terminology around head and neck cancers, oral cancers, and oropharyngeal cancers. It seems like some of these terms may be being used interchangeably. Also, suggest consistency about when these are abbreviated and when they are not.

5. How are the SNPs for each instrument combined? Was weighting used?

6. Could there be overlapping participants in the exposure and outcome consortia?

7. Minor issue, but it looks like ICD-10 codes were used, but they were simply referred to as ICD codes. Suggest specifying the ICD version.

8. Has risk-taking been used as a proxy for HPV infection before? I understand that it has been associated with sexual behaviors previously, but I'm not convinced that the evidence is strong enough to use it as the authors' have.

9. Why was 70% power chosen when 80% is standard? Why was the OR threshold used for power analyses chosen?

10. Suggest changing the phrasing for non-significant associations from "limited evidence" to "no evidence". Suggest also being clear about what "some evidence" means on line 345.

11. Suggest clarifying the sensitivity analysis section of the results. I'm not sure what these sensitivity analyses show. Do they make the same point as the main analisis?

12. I'm not sure that smoking can be concluded to be a mediator. Suggest clarifying.

---

## [Author Response]

Essential revisions:Please better explain the reasons for not including FinnGen as an outcome GWAS source. As suggested by the reviewer, 'MR could have been run using the summary statistics from FinnGen, and these MR results were then meta-analysed with those from your original cohort. ' As rightly pointed out by the reviewers, the disease outcomes are available in FinnGen, and using this strategy would increase sample size and statistical power.

We thank the reviewer for this suggestion to use FinnGen as a second outcome source. Unfortunately, following discussion with the FinnGen team, the GWAS for oral and oropharyngeal cancer are only available for release to the internal consortium (DF10) and not the current (DF8) public release. When looking at other FinnGen endpoints, such as the combined all ‘Malignant neoplasms of lip, oral cavity and pharynx’ (https://r6.risteys.finngen.fi/phenocode/C3_LIP_ORAL_PHARYNX), this includes salivary gland tumours which are often not of a squamous cell carcinoma histological subtype or nasopharyngx cases which have a different aetiology e.g., Epstein-Barr virus for nasopharyngeal cases. External lip is also considered different, given it is primarily driven by sun exposure/ ultraviolet radiation.

There is a one-year embargo on the oral and oropharyngeal cancer GWAS results, during which only the FinnGen consortium partners have access to the results. Additional cases are being collated in GAME-ON, so future replication of these analyses will be conducted when the data becomes available, but is currently not possible. UK Biobank contains a small number of n = 839 oral and oropharyngeal cancer cases, but using this for replication could result in bias, given potential sample overlap as many of the metabolic trait exposure GWAS were derived or replicated in UK Biobank.

Another issue pointed out by the reviewers is the possibility of SNPs for the metabolic traits to act pleiotropically on the other metabolic traits. In case this possibility is true, it is hard to differentiate if those SNPs are acting through the studied trait or through another mechanism.

We acknowledge concerns about potential pleiotropy in these analyses, however this was tested using a number of approaches. Evidence of heterogeneity in the effect estimates across SNPs can be assessed, which may indicate the presence of pleiotropy. The Cochran’s Q statistic was employed to detect heterogeneity, in addition to SNP outlier tests. Further pleiotropy-robust MR approaches (e.g., median and mode estimators) were also used to detect potential pleiotropy. Given the lack of effect for metabolic traits on HNC in our primary analysis, we did not correct or account for pleiotropy (e.g., using CAUSE or multivariable MR approaches). The potential for pleiotropy as a limitation has been expanded on in the main text, described in further detail below.

Also, the choice of a threshold of 0.001 for the linkage disequilibrium needs to be better justified, as this option might affect the number of SNPs related to BMI.

Clumping data at r^2^ < 0.001 is the updated default threshold used in the TwoSampleMR package to ensure potentially correlated SNPs were independently associated with the respective traits in large GWAS. Failure to account for linkage disequilibrium between SNPs can lead to overestimation of instrument strength and overly precise effect estimates. This has been further justified in main text, as described below.

Moreover, the authors could elaborate on the reasons -for using SNPs identified in from the multi-ancestry HbA1c MAGIC GWAS rather than the European-specific.

The European summary statistics for HbA_1c_ were used from the Wheeler *et al.* trans-ancestry HbA_1c_ GWAS. We have now re-written the Methods section to ensure this is clear, as described below in the response to reviewer comments.

Furthermore, the reviewers raised questions about the suitability of measuring lifetime smoking as ever vs never and using risk-taking as a proxy for HPV infection.

We have now added additional instrument-risk factor analysis using the comprehensive smoking index, a quantitative lifetime measure of smoking behaviour. We agree, there are limitations with using risk tolerance as a proxy for both sexual behaviour and HPV transmission, however genetic instruments are not available specifically for oral sex, which is the conceptually relevant exposure and likely mode of HPV transmission. We have now expanded on this in our Discussion section as described below.

Reviewer #1 (Recommendations for the authors):This is a thorough well-conducted MR analysis, just some minor points to be considered:– Make the aim clearer in the Introduction.

The study aim has now been re-written as below for clarity:

“This study aims to examine the causal effect of metabolic traits on the risk of oral and oropharyngeal cancer using two-sample MR. Specifically, we will examine adiposity measures (BMI, WC, WHR), glycaemic traits (T2D, glycated haemoglobin (HbA_1c_), fasting glucose (FG), fasting insulin (FI)), and blood pressure (SBP, DBP).” (page 6, lines 117-121)

– Was there any reason why FinnGen was not included as an outcome GWAS source? MR could have been run using the summary statistics from FinnGen and these MR results were then meta-analysed with those from your original cohort (which doesn't appear to contain FinnGen or have any risk of overlap). Both disease outcomes exist in FinnGen, and while there are few cases of oropharyngeal cancer, there are many for oral cancer, which would greatly increase your sample size and thus increase power. Unless there is a specific reason not to include FinnGen, I recommend that the analysis should be re-done (at least as a sensitivity analysis) including the publicly available FinnGen summary statistics.

We thank the reviewer for this suggestion to use FinnGen as a second outcome source. Unfortunately, following discussion with the FinnGen team, the GWAS for oral and oropharyngeal cancer are only available for release to the internal consortium (DF10) and not the current (DF8) public release. When looking at other FinnGen endpoints, such as the combined all ‘Malignant neoplasms of lip, oral cavity and pharynx’ (https://r6.risteys.finngen.fi/phenocode/C3_LIP_ORAL_PHARYNX), this includes salivary gland tumours which are often not of a squamous cell carcinoma histological subtype or nasopharyngx cases which have a different aetiology e.g., Epstein-Barr virus for nasopharyngeal cases. External lip is also considered different, given it is primarily driven by sun exposure/ ultraviolet radiation.

There is a one-year embargo on the oral and oropharyngeal cancer GWAS results, during which only the FinnGen consortium partners have access to the results. Additional cases are being collated in GAME-ON, so future replication of these analyses will be conducted when the data becomes available, but is currently not possible. UK Biobank contains a small number of n = 839 oral and oropharyngeal cancer cases, but using this for replication could result in bias, given potential sample overlap as many of the metabolic trait exposure GWAS were derived or replicated in UK Biobank.

– "A meta-analysis of observational studies investigating T2D with oral and oropharyngeal subsites, showed an increased risk ratio (RR) of 1.15, 95%CI 1.02-1.29, P heterogeneity = 0.277" – did you mean to give only the P-value for the heterogeneity test here? Seems like the P-value relating to the RR should also be included.

This was the overall random-effects meta-analysis result which is why P heterogeneity was given. On calculation using 95%CI and the effect estimate, *p* is <0.001. This has now been updated as below:

“A random-effects meta-analysis of observational studies showed an increased association between T2D and oral and oropharyngeal cancer risk ratio (RR) of 1.15, 95%CI 1.02–1.29, p<0.001 (Gong, Wei, Yu, & Pan, 2015).” (page 4, lines 80-83)

– "investigate if an exposure (X), is associated with a trait e.g., body mass index or disease outcome (Y)" – this sentence is misleading to the reader as you have used BMI as an exposure (Figure 1), so this sentence should be re-phrased.

Thank you for highlighting this. We have now re-phrased the description of **Figure 1** to ensure clarity, as below:

“Genetic variants (G) can act as proxies or instruments to investigate if an exposure (X) is associated with a disease outcome (Y). Causal inference can be made between X and Y if the following conditions are upheld:

(1) The genetic variants which make up the instrument are valid and reliably associated with the exposure (i.e., the ‘relevance assumption’);

(2) There is no measured or unmeasured confounding of the association between the genetic instrument and the outcome (i.e., the ‘exchangeability’ assumption);

(3) There is no independent pathway between the genetic instrument and the outcome, except through the exposure (i.e., the ‘exclusion restriction principle’).”

(Figure 1, pages 5-6, lines 103-113)

– Your linkage disequilibrium (R2) threshold of 0.001 seems very low – was there any specific reason behind this choice of threshold? Maybe briefly explain the reasoning behind this choice in the text. As a result, you end up with few variants – for example, you only have 312 SNPs for BMI – other studies using BMI as an exposure in MR analyses have used ~700 variants from the GIANT consortium.

Clumping data at r^2^ < 0.001 is the updated default threshold used in the TwoSampleMR package (https://mrcieu.github.io/TwoSampleMR/reference/clump_data.html) to ensure potentially correlated SNPs were independently associated with the respective traits in large GWAS. Failure to account for linkage disequilibrium between SNPs can lead to overestimation of instrument strength and overly precise effect estimates. This has been further justified in main text as below:

“Clumping was performed in the TwoSampleMR package to ensure single nucleotide polymorphisms (SNPs) in each instrument were independent (r^2^ < 0.001). This accounted for any potential linkage disequilibrium between SNPs, which can lead to overestimation of instrument strength and overly precise effect estimates.” (page 19, lines 371-375)

– When you describe the MAGIC cohort you do not mention the ancestry, as you did with the other cohorts – please also include this.

This has now been updated to include individuals of European descent for the MAGIC Consortium as below:

“..33 SNPs for FG and 18 SNPs for FI, obtained from a GWAS published by the MAGIC (Meta-Analyses of Glucose and Insulin-Related Traits) Consortium (N = 151,188 and 105,056 individuals of European descent, respectively) (Lagou et al., 2021)…” (page 20, lines 383-385)

– Why did you use SNPs identified in the multi-ancestry HbA1c MAGIC GWAS rather than the European-specific one when this is available?

Thank you for highlighting this. The European summary statistics for HbA_1c_ were used from the Wheeler *et al.* study. We have now re-written the Methods section as below to ensure this is clear as below:

“58 SNPs for HbA_1c_, taken from a meta-analysis of 88,355 individuals from European cohorts (Wheeler et al., 2017)…” (page 20, lines 385-386)

– Did you use proxy SNPs when your genetic variants of interest were not available in the outcome GWAS summary statistics? If so, how were these proxy SNPs identified in terms of R2 thresholds for LD with the index variant, distance from the index variant, data sources used, etc.?

Proxy SNPs weren’t used as most of the variants were lost from the instrument due to initial clumping which we hope has been justified above.

– "We only used genetic variants reaching GWAS significance (p <5x10-8)." – you should explicitly state that this is only for the exposure, and not the outcome.

This statement has now been rephrased as below:

“For exposures, we only used genetic variants reaching GWAS significance (p <5×10^−8^).” (page 21, lines 423-424)

– "individual had ever smoked in their life versus never smokers" – is this the best smoking measure? Surely a variable that measures whether someone was a regular smoker or not would be better if available – maybe expand on this in limitations.

As well as investigating the effects of smoking initiation, we have now added a quantitative lifetime measure of smoking behaviour using the comprehensive smoking index, as described below:

“The comprehensive smoking index, a quantitative lifetime measure of smoking behaviour derived from 462,690 individuals from UK Biobank was also employed. A 1 standard deviation (SD) increase in the index is equivalent to an individual smoking 20 cigarettes a day for 15 years and stopping 17 years ago, or an individual smoking 60 cigarettes a day for 13 years and stopping 22 years ago.” (page 24, lines 479-483)

“Smaller, yet similar effects were found between adiposity measures and the comprehensive smoking index: BMI (Β IVW 0.10 (standard error (SE) 0.01), p < 0.001, per 1 SD increase in BMI (4.81 kg/m2)), WC (Β IVW 0.10 (SE 0.02), p < 0.001, per 1 SD increase in WC (0.09 unit)) and WHR (Β IVW 0.09 (SE 0.01), p < 0.001, per 1 SD increase in WHR (0.10 unit))” (page 13, lines 232-235)

– You state that there are known strong genetic correlations between risk-taking and sexual behavior – however, it still may not be a great proxy for the latter – maybe state this in your limitations.

We agree, there are limitations with using risk tolerance as a proxy for both sexual behaviour and HPV transmission and have now expanded on this in our Discussion section, as below:

“Risk tolerance is challenging to instrument genetically due to measurement error (e.g., as a result of reporting bias) and because it is socially patterned, time-varying as well as context and culture-dependent (Gormley et al. 2022). It may also be a poor proxy for sexual behaviour, despite genetic correlation with these phenotypes given that pleiotropy with other traits such as smoking may be present (Mills et al. 2021). However, genetic instruments are not available specifically for oral sex, which is the conceptually relevant exposure and likely mode of HPV transmission.” (page 18, lines 335-341)

– "SNPs used to proxy these metabolic traits, particularly obesity-related measures BMI, WC, and WHR were strongly associated with smoking." – you mention this very briefly at the end, and expand on this potential limitation.

This limitation has now been expanded on as below:

“SNPs used to proxy these metabolic traits, particularly adiposity measures BMI, WC and WHR were also strongly associated with smoking. Repeating this analysis in an updated, better powered GWAS is required in order to exclude any potential small effects of metabolic traits on HNC risk via smoking. Given the heterogeneity of these complex metabolic traits, future work could further examine their pathway specific effects (Udler et al., 2018).” (page 18, lines 343-348)

– "Although there is no clear evidence that changing body mass will reduce of increase the risk" – 'or' instead of 'of'?

This typo has now been corrected as shown below:

“Although there is no clear evidence that changing body mass will reduce or increase the risk of HNC directly…” (page 18, lines 357-358)

Reviewer #2 (Recommendations for the authors):1. Do any of the SNPs for the metabolic traits act pleiotropically on the other metabolic traits? If so, it would be hard to know whether those SNPs are acting through the studied trait or through another mechanism (like another correlated metabolic trait).

We acknowledge the reviewer’s concerns about potential pleiotropy in these analyses, however this was tested using a number of approaches. Evidence of heterogeneity in the effect estimates across SNPs can be assessed, which may indicate the presence of pleiotropy. The Cochran’s Q statistic was employed to detect heterogeneity, in addition to SNP outlier tests. Further pleiotropy-robust MR approaches (e.g., median and mode estimators) were also used to detect potential pleiotropy. Given the lack of effect for metabolic traits on HNC in our primary analysis, we did not correct or account for pleiotropy (e.g., using CAUSE or multivariable MR approaches).

However, this limitation, particularly with respect to the analysis of HNC risk factors on metabolic traits has now been expanded on as below:

“SNPs used to proxy these metabolic traits, particularly adiposity measures BMI, WC and WHR were also strongly associated with smoking. Repeating this analysis in an updated, better powered GWAS is required in order to exclude any potential small effects of metabolic traits on HNC risk via smoking. Given the heterogeneity of these complex metabolic traits, future work could further examine their pathway specific effects (Udler et al., 2018).” (page 18, lines 343-348)

2. Is there any way to replicate these findings?

As discussed above in response to the first reviewer’s comments, there is a one-year embargo during which only the FinnGen consortium partners have access to the results. Additional cases are being collated in GAME-ON, so future replication of these analyses will be conducted when the data becomes available, but is currently not possible. UK Biobank contains a small number of n = 839 oral and oropharyngeal cancer cases, but using this for replication could result in bias, given potential sample overlap as many of the metabolic trait exposure GWAS were derived or replicated in UK Biobank.

3. It is unclear how the last sentence of the abstract can be used as a conclusion based on the results portion of the abstract. Suggest revising.

This sentence has now been removed given the limited space to explain this in the results portion of the abstract.

4. Suggest clearly using terminology around head and neck cancers, oral cancers, and oropharyngeal cancers. It seems like some of these terms may be being used interchangeably. Also, suggest consistency about when these are abbreviated and when they are not.

Head and neck cancer has now been abbreviated to HNC, with oral and oropharyngeal cancer subsites written in full throughout the paper. Alternative terms such as oropharynx have been changed to oropharyngeal for consistency.

5. How are the SNPs for each instrument combined? Was weighting used?

The weighting used in the two-sample MR method is described in the Methods section, as below:

“For each SNP in each exposure, individual MR effect-estimates were calculated using the Wald method (SNP-outcome β/SNP-exposure β) (Wald, 1940). Multiple SNPs were then combined into multi-allelic instruments using random-effects inverse-variance weighted (IVW) meta-analysis.” (page 22, 431-434)

6. Could there be overlapping participants in the exposure and outcome consortia?

It is unlikely there would be any sample overlap, as the GAME-ON outcome data is made up of 12 independent head and neck cancer specific cohorts from the US, Europe and South America. None of the exposure consortia were included in the outcome GWAS, including UK Biobank. Further detail on these head and neck cancer studies can be found in **Table 1** of the dpGaP OncoArray: Oral and Pharynx Cancer webpage: https://www.ncbi.nlm.nih.gov/projects/gap/cgi-bin/study.cgi?study_id=phs001202.v1.p1

7. Minor issue, but it looks like ICD-10 codes were used, but they were simply referred to as ICD codes. Suggest specifying the ICD version.

Thanks for highlighting this, which has now been changed to: “ICD-10” (page 20, line 402)

8. Has risk-taking been used as a proxy for HPV infection before? I understand that it has been associated with sexual behaviors previously, but I'm not convinced that the evidence is strong enough to use it as the authors' have.

As above, we agree there are limitations with using risk tolerance as a proxy for both sexual behaviour and HPV transmission and have now expanded on this in our Discussion section as below:

“Risk tolerance is challenging to instrument genetically due to measurement error (e.g., as a result of reporting bias) and because it is socially patterned, time-varying as well as context and culture-dependent (Gormley et al. 2022). It may also be a poor proxy for sexual behaviour, despite genetic correlation with these phenotypes given that pleiotropy with other traits such as smoking may be present (Mills et al. 2021). However, genetic instruments are not available specifically for oral sex, which is the conceptually relevant exposure and likely mode of HPV transmission.” (page 18, lines 335-341)

9. Why was 70% power chosen when 80% is standard? Why was the OR threshold used for power analyses chosen?

We agree with the reviewer and power has now been recalculated at 80%, with Figure 2—figure supplement 1 also updated. This did not change the interpretation of results as noted below:

“In analyses where BMI was the exposure, we had 80% power to detect an association with an OR of 1.2 or more at an α of 0.05 for combined oral and oropharyngeal cancer.” (page 7, lines 133-135)

10. Suggest changing the phrasing for non-significant associations from "limited evidence" to "no evidence". Suggest also being clear about what "some evidence" means on line 345.

We take on board the reviewer’s comments but disagree on changing the term “limited evidence” for “no evidence”, as absence of evidence for an effect does not mean there isn’t any. It is never reasonable to claim that a study has proved no effect, as some uncertainty will always exist in any study design given their individual limitations. Although our study contributes evidence, future replication and triangulation of study methods is required to answer this question.

We agree the term “some evidence” is not clear and have therefore removed this as below:

“Where there was evidence for an effect…” (page 13, line 225)

11. Suggest clarifying the sensitivity analysis section of the results. I'm not sure what these sensitivity analyses show. Do they make the same point as the main analisis?

As noted above in response to the second reviewer’s point 1, these sensitivity analyses were performed to explore the presence of heterogeneity or genetic pleiotropy. The results of these sensitivity analyses generally followed the same pattern as the IVW results reported above, and where there were exceptions, further outlier analysis and SIMEX correction was performed.

12. I'm not sure that smoking can be concluded to be a mediator. Suggest clarifying.

Regarding the potential for smoking to act as a mediator, in reality it's possible that it could be acting as both a mediator and a confounder since the relationship between BMI and smoking is bi-directional (i.e., smoking reduces BMI and higher BMI in turn increases likelihood of smoking). This has been shown in previous MR studies (Carreras-Torres et al. 2018; Taylor et al. 2018) and has been added to the Discussion section for clarity as below:

“Smoking could be acting as both a mediator and a confounder, since the relationship between BMI and smoking is bi-directional (i.e., smoking reduces BMI and higher BMI in turn increases the likelihood of smoking), which has been demonstrated in previous MR studies (Carreras-Torres et al. 2018; Taylor et al. 2018).” (page 16, lines 290-294).

References

Carreras-Torres R, Johansson M, Haycock PC, Relton CL, Davey Smith G, Brennan P, Martin RM. 2018. Role of obesity in smoking behaviour: Mendelian randomisation study in UK Biobank. BMJ (Clinical research ed). 361:k1767-k1767.

Gormley M, Dudding T, Kachuri L, Burrows K, Chong AHW, Martin RM, Thomas SJ, Tyrrell J, Ness AR, Brennan P et al. 2022. Investigating the effect of sexual behaviour on oropharyngeal cancer risk: a methodological assessment of Mendelian randomization. BMC Medicine. 20(1):40.

Mills MC, Tropf FC, Brazel DM, van Zuydam N, Vaez A, Pers TH, Snieder H, Perry JRB, Ong KK, den Hoed M et al. 2021. Identification of 371 genetic variants for age at first sex and birth linked to externalising behaviour. Nat Hum Behav. 5(12):1717-1730.

Taylor AE, Richmond RC, Palviainen T, Loukola A, Wootton RE, Kaprio J, Relton CL, Davey Smith G, Munafò MR. 2018. The effect of body mass index on smoking behaviour and nicotine metabolism: a Mendelian randomization study. Hum Mol Genet. 28(8):1322-1330.